# Pelvic Venous Insufficiency: Input of Short Tau Inversion Recovery Sequence

**DOI:** 10.3390/jpm12122055

**Published:** 2022-12-13

**Authors:** Eva Jambon, Yann Le Bras, Gregoire Cazalas, Nicolas Grenier, Clement Marcelin

**Affiliations:** Department of Radiology, Pellegrin Hospital, Place Amélie Raba Léon, 33076 Bordeaux, France

**Keywords:** magnetic resonance imaging, pelvis, venous congestion, phlebography, varicose veins, venous insufficiency

## Abstract

**Objectives:** To evaluate indirect criteria of pelvic venous insufficiency (PVI) of a short tau inversion recovery (STIR) sequence retrospectively compared with phlebographic findings. **Methods:** Between 2008 and 2018, 164 women who had received MRI and phlebography for pelvic congestion syndrome (60), varicose veins in the lower limbs (45), both (43), or other symptoms (16) were included. The presence of periuterine varicosities and perivaginal varicosities were compared to the findings of phlebography: grading of left ovarian vein reflux and presence of internal pudendal or obturator leak. **Results:** There was a correlation between the grading of LOV reflux on phlebography and the diameter of periuterine varicosities on STIR sequence (*p* = 0.008, rho = 0.206, CIrho [0.0549 to 0.349]). Periuterine varicosities had a positive predictive value of 93% for left ovarian reflux (95% CI [88.84% to 95.50%]). Obturator or internal pudendal leaks were found for 118 women (72%) and iliac insufficiency for 120 women (73%). **Conclusions:** Non-injected MRI offers a satisfactory exploration of PVI with STIR sequence. STIR sequences alone enabled the detection of left ovarian and iliac insufficiency.

## 1. Introduction

Pelvic venous insufficiency (PVI) is a pathology of premenopausal and multiparous women that is still poorly understood and frequently misdiagnosed [1]. Regularly responsible for pelvic congestion syndrome (PCS), combining chronic pelvic pain, dysmenorrhea, and dyspareunia, PVI induces recurrent varicose veins in the lower limbs (VVLL) and unsightly varicosities on the buttock, perineal, or vulvar [2,3,4]. These interrelated symptoms were recently grouped under the term “Pelvic Venous Disorders” [5]. The diversity of symptoms and the poor understanding of pathogenic mechanisms are major concerns. Although first described by Taylor in 1949, PCS frequency remains unknown [6,7]. The prevalence of retrograde flow in the ovarian veins has been estimated at 9.9% based on 273 angiographic studies including healthy female renal transplant donors, with 59% of them reporting chronic pelvic pain compatible with PCS and 77% reporting improvement after ovarian vein ligation [8,9]. A recent study on 2384 abdomino-pelvic computed tomography (CT) images found that 8% of premenopausal women had PCS with dilated left ovarian veins (LOV) [10].

Non-invasive investigations are recommended as an initial assessment for PVI, but phlebography remains the gold standard with a sensitivity and specificity of 91% and 89%, respectively [11,12,13]. However, phlebography is an invasive diagnostic technique, exposes the pelvis of women of childbearing age to irradiation, is costly, and is time-consuming [14,15]. Thanks to non-invasive diagnostic tools, pre-therapeutic phlebography could be facilitated [15].

CT is an effective technique for this indication but it is irradiating [16,17,18,19,20]. MRI is the imaging modality of choice for female pelvic exploration with an optimal exploration of internal genital organs [21] to eliminate differential diagnoses of chronic pain, in particular, endometriosis with similar symptoms [15,22].

Many women with PVI had an important diagnostic delay, sometimes for many years. A non-injected sequence could be added to MRI for “undetermined chronic pelvic pain” suitable for PVI.

Short-tau inversion recovery (STIR) sequence is an IR technique that nulls fat signal intensity based on T1 values. The STIR sequence uses an initial 180° RF pulse to invert spins in the longitudinal plane. After a short time delay (known as inversion time (TI), this initial RF pulse is followed by a conventional spin-echo. To achieve fat suppression, a TI should be selected such that the longitudinal magnetization of the fat spins is zero when a subsequent 90° pulse is applied. The TI that will negate the signal from fat is equal to 0.69 times the Ti relaxation time of fat, provided that the selected TA is significantly greater than Ti. As Ti relaxation times are proportional to the applied magnetic field, the appropriate TI with a STIR for nulling the signal from fat must be adjusted for a given magnet strength. Even at a given magnetic field strength, the TI that maximally nulls the fat signal varies slightly from patient to patient, possibly because of differences in fat composition. STIR pulse sequences rely on the relatively short Ti relaxation time of adipose tissue to achieve fat suppression. This technique is quite different from the frequency-selective fat suppression technique, in which the signal from fat protons is selectively nulled on the basis of the intrinsic chemical shift differences between lipid and water protons. As STIR sequences will suppress the signal from any tissue or fluid that, like fat, has a short Ti relaxation time, this technique of fat suppression may be considered nonselective [23]. The STIR sequence allows unparalleled cartography of the pelvis. It is not yet evaluated despite its excellent tissue contrast and spatial resolution. PVI induces important varicosities visualized like hyperintense dilated tortuous structures around the uterus and vagina.

The aim of this study was to evaluate the indirect criteria of PVI of STIR sequences retrospectively compared with phlebographic findings.

## 2. Material and Methods

### 2.1. Patients

This was a single-centre, retrospective study performed in a regional University Hospital. Ethical approval was obtained by the Publication Group of the Ethics committee of the University Hospital (CE-GP-2019-20). Using radiology databases and medical records at our institution, from 2008 to 2018, we consecutively included all women who had MRI and phlebography for suspicion of PVI based on clinical features, defined as chronic pelvic pain (>6 months) with dysmenorrhea, dyspareunia, or post-coital pain, that typically increased at the end of the day and while standing. Other signs could be associated with dysuria, nocturia, or rectal symptoms. VVLL included were evaluated by US and pelvic leaks were identified. Unsightly varicosities on the buttock, perineal, or vulvar were described by the patients and clinically observed. A pelvic leak was defined as an abnormal communication between VVLL and pelvic veins. Exclusion criteria were other aetiologies of chronic pelvic pain (uterine fibroma, adenomyosis, or endometriosis).

### 2.2. Magnetic Resonance Imaging

Imaging was performed on a 1.5 Tesla instrument (Avanto, Siemens Healthcare, Forchheim, Germany) The duration of the examination was 30 to 40 min. After the acquisition of the scout images, 2-dimensional T2-weighted STIR BLADE sagittal and coronal images were acquired as the detailed scout images. The acquisition parameters for STIR images were as follows: TR, 4810 ms; TE, 82 ms; matrix, 256 × 256; slices, 30; slice thickness/gap, 4.5 mm/10% and imaging time, 3 min 22 s. Then, 2-dimensional T2-weighted STIR BLADE axial images were acquired. The acquisition parameters for axial STIR images were as follows: TR, 5771 ms; TE, 82 ms; matrix, 256 × 256; slices, 35; slice thickness/gap, 4.5 mm/10% and imaging time, 4 min 02 s. After, the other sequences were realized: axial T2 TRUFI, Axial T1 DIXON 3D VIBE, Angio Dynamic TWIST with coronal MIP reconstruction then 3D T1 VIBE. Dotarem© was injected at 3 mL/sec at the third dynamic for 2 mL/kg and flushed with 20 mL of NaCl.

A differential diagnosis or a type II or III venous compression, according to the classification of Greiner et al., had to be eliminated [24]. Indirect criteria of PVI of STIR in the 3 spatial planes with visualization of the top of the thighs were obtained to record the diameter of periuterine veins and the presence of perivaginal or external varicosities. A diameter > 5 mm was considered a periuterine varicosity, according to the Guidelines of the Society of Interventional Radiology [12]. The presence of paravaginal, perineal, vulvar varicosities or varicosities on the buttock was defined by abnormal dilated and tortuous veins. The presence of a hyposignal STIR sequence within pelvic veins was noted like “flow voids”.

### 2.3. Phlebography

All procedures were undertaken on the same angiographic unit (Philips Medical Systems). Phlebography was performed by two investigators with experience in interventional radiology of 5 and 20 years before the embolization. A brachial or femoral venous approach was used, using a 4- or 5-French sheath. LRV was catheterized with a 4-French Cobra catheter and phlebography was performed. If a Nutcracker syndrome was suspected with clinical symptoms (haematuria, lumbar pain), a measurement of the vena cava and renal vein pressures was realized (Normal ≤ 3 mmHg) [24]. Then, LOV was catheterized and a phlebography with Valsalva manoeuvre was realized with automatic injection (Visipaque^®^: volume of 20 mL, rate of 10 mL/s). The internal iliac veins were catheterized with a Cobra catheter or a UAC catheter and were looking for incompetent collectors, leaks to the legs, and a May-Thurner syndrome with automatic injection (Visipaque^®^: volume of 20 mL, rate of 10 mL/s).

LOV insufficiency was defined by reflux towards the pelvis. LOV diameter was noticed and reflux was scored from 0 to 3 according to the description by Hiromura et al. [25]. In grade 1, retrograde flow remained in the LOV, not reaching the parauterine veins. In grade 2, the retrograde flow advanced into the ipsilateral parauterine veins and no further. In grade 3, the retrograde flow crossed the midline passing through the uterus from the left into the right parauterine plexus [25].

Iliac insufficiency was defined by reflux into its tributaries. A superior gluteal leak was visualized in front of the iliac wing, an inferior gluteal leak in front of the femoral head, an obturator leak passed over the obturator foramen and a pudendal leak followed the iliopubic rami. Irradiation data were not relevant because the embolization was performed on the same day.

### 2.4. Imaging Analysis

Images were reviewed in a blinded fashion by one radiologist with five years of experience and one radiologist with twenty years of experience. Inter-observer variability was evaluated for dilatation of iliac tributaries. In case of disagreement, a consensus reading was organized for a final decision. Only clinical patient information was available. To avoid any recognition bias, the studies were presented in random order. All imaging investigation data were assessed on a Picture Archiving and Communication system (PACS) station (Vue PACS; Carestream Health, Rochester, NY, USA).

### 2.5. Statistical Analysis

Statistical analysis was performed with Excel and MedCalc^®^ software version 12.3 (Ostend, Belgium). Qualitative variables are expressed as raw numbers, proportions, and percentages. Quantitative variables are expressed as medians with 1st (Q1) and 3rd (Q3) quartiles and ranges.

The positive predictive value was calculated for every criterion. The correlation between reflux grading and periuterine varicosities was tested using Spearman’s rank correlation. A *p*-value < 0.05 was considered statistically significant.

The inter-observer variability coefficient was calculated by the Kappa Cohen coefficient.

## 3. Results

### 3.1. Patients and Anatomical Analysis

Two hundred and twenty-five women had a phlebography for suspicion of PVI between 2008 and 2018 at our centre. One hundred and sixty-four women (median age: 39 years; Q1–Q3: 34–45) were included in the study (Table 1): 60 women for a PCS, 45 for VVLL, 43 for both, and 16 for other isolated symptoms (6 women with vulvar varicosities, 4 patients with perineal varicosities, 1 patient with varicosities on the buttock and 5 women suffering with lumbar pain and haematuria were included with suspicion of Nutcracker syndrome).

Referrals were made by vascular specialists in 62% of cases, vascular surgeons in 26%, and gynaecologists in 12%. The median time between MRI and phlebography was 41 days (28–65). One patient had an angioedema caused by a Gadolinium injection that required a short stay in intensive care.

Anatomical variability was found for 16 women (10%). Seven patients had LOV variability: six women with two LOV (4%) and one woman with three. There were five patients with LRV variability: one woman with two LRV, three with retroaortic LRV (2%), and one with circumaortic. There were three ROV draining into the right renal vein (2%).

### 3.2. STIR and Phlebographic Findings

On the STIR sequence, the median diameter of periuterine veins was 7 mm (6–8). One hundred and twenty-six women had periuterine varicosities with a diameter > 5 mm (77%). Ninety-eight patients had “flow voids” in their pelvic varicosities (60%).

One hundred and twenty-nine patients had perivaginal varicosities (79%), thirty-nine vulvar varicosities (24%), fifty-seven perineal varicosities (35%), and five varicosities on the buttock (3%).

One hundred and forty-four women had left ovarian reflux on phlebography (88%). Six reflux were grade 1 (4%), thirty-nine reflux were grade 2 (27%), and ninety-nine were grade 3 (69%).

There was a correlation between the grading of LOV reflux on phlebography and the diameter of periuterine varicosities on the STIR sequence (*p* = 0.008, rho = 0.206, CIrho [0.0549 to 0.349]). Periuterine varicosities had a positive predictive value of 93% for left ovarian reflux (95% CI [88.84% to 95.50%]).

Obturator or internal pudendal leaks were found for 118 women (72%) and iliac insufficiency for 120 women (73%) (Figure 1). Inter-observer agreement for iliac insufficiency was substantial (k = 0.72).

Perivaginal varicosities had a positive predictive value of 78% for obturator or internal pudendal leak (95% CI [64.41% to 78.68%]) (Figure 2). Perivaginal or external varicosities had a predictive value of 76% for iliac insufficiency (95% CI [72.51% to 79.71%]) (Table 2).

## 4. Discussion

Based on 10 years of experience reported here, these data confirm that the STIR sequence is indirectly reliable to detect PVI. Detection of periuterine varicosities had a positive predictive value of 82% for LOV reflux. This result was according to the transvaginal US findings in the literature relating a sensitivity and specificity of 100% and 83–100%, respectively [26,27]. Pelvic US had many advantages: localization of pain at the passage of the probe and use of a Valsalva maneuver to objective reflux. US had the main disadvantage of being direct and operator-dependent. The STIR sequence allows for realistic cartography of pelvic varicosities for optimal anticipation and processing of the procedure of embolization. The iliac, ovarian, or mixt supply is visualized with the facility.

The presence of perivaginal varicosities had a 78% positive predictive value for obturator or pudendal leaks. The composite criterion containing peri-vaginal and external varicosities detected on the STIR sequence had a positive predicted value of 76% for iliac insufficiency.

In the literature, the evaluation of PVI was limited to direct reflux analysis [9,14,28,29]. Asciutto et al. evaluated dynamic MRI versus phlebography in 23 patients with PCS. The sensitivity and specificity of MRI were, respectively, 88% and 67% for ovarian veins, 100% and 38% for hypogastric veins, and 91% and 42% for periuterine varicosity [14]. Yang et al. compared the grading of reflux in the ovarian veins on dynamic MRI versus phlebography in 19 patients: the sensitivity, specificity, and diagnosis relevance of MRI were 66.7%, 100%, 78.9%, and 75%, 100%, 84.2% for the two observers, respectively [29]. Meneses et al. used a phase-contrast velocity mapping and found a sensitivity of 100% and specificity of 50%, based on nine patients with suspected PCS [30].

Perivaginal varicosities were responsible for deep dyspareunia and post-coital pains. Any radiologists could detect these varicosities with an endovaginal probe, with the copying of usual pains [31]. The sagittal STIR sequence allows complete visualization of peri-vaginal varicosities and these venous supplies. This criterion could be used to facilitate the phlebography and to search for internal pudendal or obturator incompetency even if the vein is valvulated.

External varicosities were visualized very well by the STIR sequence like a serpiginous dilated hypersignal vein in subcutaneous tissue. These veins were often in communication with varicose in the lower limbs. They could be treated directly by sclerotherapy with 2% polidocanol (Aetoxisclerol, Kreussler, Wiesbaden, Hessen, Germany) under ultrasound and X-ray guidance [32].

It is well known that PVI was a complex pathology and the ovarian vein reflux analysis is not sufficient. The dynamic MRI sequence does not allow for analyzing the iliac vein and afferents reflux [14]. STIR sequence allows a direct and indirect analysis of ovarian and iliac veins with a good spatial resolution.

Our study has some limitations, notably with recruitment, as all symptomatic patients had a final diagnosis of PVI. No patient had an MRI or phlebography that did not result in a PVI diagnosis. The retrospective review of imaging examinations could lead to the recognition of records. We have tried to minimize this bias by presenting the studies in random order.

In conclusion, non-injected MRI offers a satisfactory exploration of symptomatic PVI with the STIR sequence. Indirectly, the STIR sequence alone enabled the detection of left ovarian and iliac insufficiency.

Gynecological imaging included 3D T2 weight or sagittal T2 weight, however, these sequences provide detection of anomalies in pelvic organs, but the pelvic fat avoids correctly detecting varicose veins or they are likely sub-estimated.

The STIR sequence could be added as an option after a usual MRI protocol performed for unexplained pelvic pain. After detection of symptomatic pelvic varicosities without other etiology of pelvic pain on MRI, the patient should be addressed to an interventional radiologist.

## Figures and Tables

**Figure 1 jpm-12-02055-f001:**
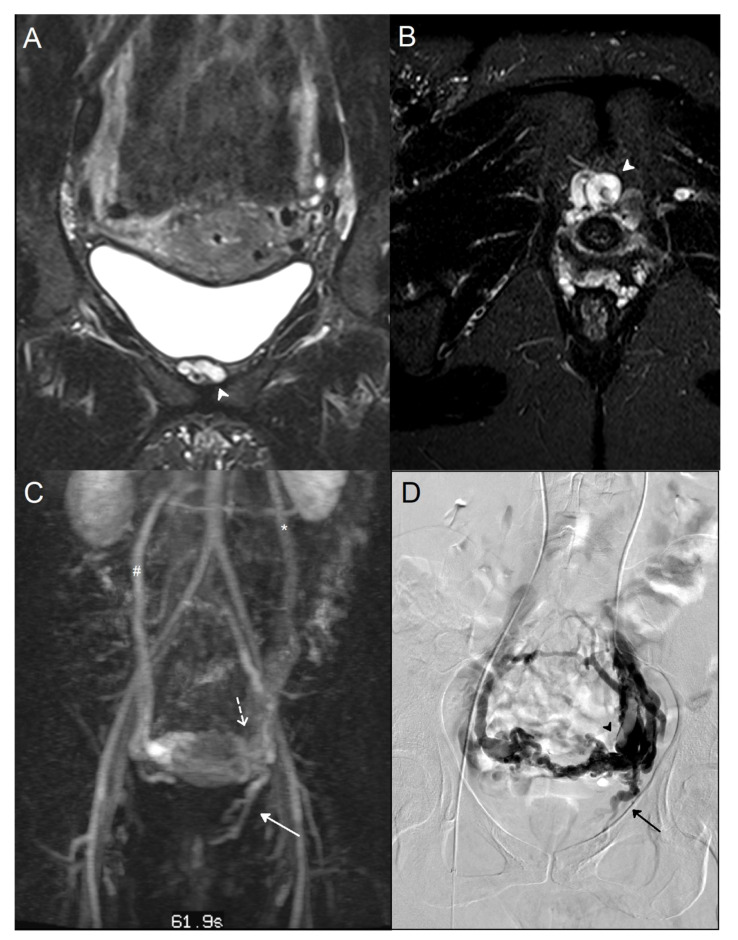
A 38-year-old woman who suffered from recurrent varicose veins in the lower limbs had magnetic resonance imaging (MRI) and a phlebography for suspicion of pelvic venous insufficiency. A/B: MRI coronal (**A**) and axial (**B**) T2 short inversion time inversion recovery (STIR) sequence: Dilated left internal pudendal vein *(white arrowhead)*. (**C**) Maximal intensity projection (MIP) coronal reconstruction of MRI angiography sequence: Voluminous periuterine varicose *(dotted arrow)*, reflux in the left internal pudendal vein *(white arrow),* incompetent left ovarian vein *(*)*, and vicarious right ovarian vein *(#)*. (**D**) Phlebography: Confirmation of an incompetent left internal pudendal vein *(black arrow).* Visualization of important periuterine varicose *(black arrowhead)*.

**Figure 2 jpm-12-02055-f002:**
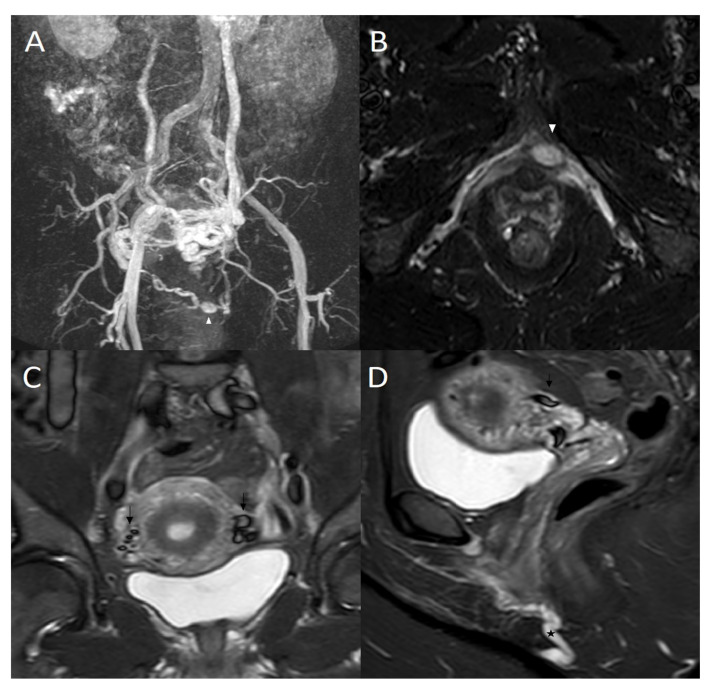
(**A**) 50-year-old woman who suffered from pelvic pain and vulvar varicosities had magnetic resonance imaging (MRI). Maximal intensity projection (MIP) coronal reconstruction of MRI angiography sequence: Voluminous left periuterine varicose with reflux in the internal pudendal veins *(white arrowhead),* incompetent left ovarian vein, and vicarious right ovarian vein. (**B**) MRI axial T2 short inversion time inversion recovery (STIR) sequence: Dilated left internal pudendal vein with aneurysm *(white arrowhead).* (**C**) MRI coronal T2 short inversion time inversion recovery (STIR) sequence: Flow voids in dilated pelvic varicose veins *(dark arrow)*. (**D**) MRI sagittal T2 short inversion time inversion recovery (STIR) sequence: Flow voids in dilated pelvic varicose veins *(dark arrow)* and dilated left internal pudendal vein *(*)*.

**Table 1 jpm-12-02055-t001:** Patients’ characteristics. All quantitative variables are expressed in mean, SD, and range/median (Q1–Q3); all qualitative variables are expressed with raw numbers, proportions, and percentages.

	164
Age (years)	39 ± 9 (21–69)/39 (34–45)
Number of pregnancies	2 ± 1 (0–6)/2 (2–3)
PCS	60 (60/164; 36.59%)
VVLL	45 (45/164; 27.44%)
PCS + VVLL	43 (43/164; 26.22%)
Presence of external varicosities * Isolated	34 (34/164; 20.73%) 16 (16/164; 9.76%)
Dysuria	4 (4/164; 2.44%)

* Vulvar, perineal or on the buttock, isolated, or in association. N: Number of patients, PCS: pelvic congestion syndrome, VVLL: varicose veins of the lower limbs.

**Table 2 jpm-12-02055-t002:** Diagnostic performance of the STIR sequence.

Criterion	Sensitivity	Specificity	PPV	NPV
Detection of leaks	0.92	0.77	0.93	0.73
Iliac insufficiency by detection of leaks	0.91	0.27	0.68	0.63
Left ovarian insufficiency by detection of Periuterine varicosities > 5 mm	0.82	0.53	0.93	0.26
Internal pudendal or obturator leak by detection of perivaginal varicosities	0.87	0.56	0.78	0.62

PPV: positive predictive value; NPV: negative predictive value.

## Data Availability

Not applicable.

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
