# Peer review of "Pelvic Venous Insufficiency: Input of Short Tau Inversion Recovery Sequence"

_jpm, 2022, doi:10.3390/jpm12122055_

Round 1

Reviewer 1 Report

To:

Editorial Board

Journal of Personalized Medicine

Title: “Pelvic venous insufficiency: Input of short tau inversion recovery sequence”

Dear Editor,

I evaluated this paper and I think that:

-       Acronyms should be expressed at their first mention both in the abstract and in the main text. Please revise the entire paper.

-       The English of the paper should be revised by a native English speaker in order to improve the readability of the text. Please provide.

-       The retrospective nature of this paper is a limitation. Please discuss such a point in a dedicated limitation section.

-       Inter/intraobserver variability coefficients should be calculated for imaging evaluations. Please provide.

-       A table gathering the main characteristics of the study population is lacking. Please provide.

-       The comparisons and the reproducibility of the methods are lacking. No table/figure had been provided dealing with the statistical comparisons of the methods.

Author Response

-       Inter/intraobserver variability coefficients should be calculated for imaging evaluations. Please provide.

Authors: A Kappa coefficient was calculated for interobserver agreement for iliac insufficiency. For others characteristics there was no disagreement.

No intraobserver agreement was calculated.

- A table gathering the main characteristics of the study population is lacking. Please provide.

Authors: The Table 1 was added accordingly.

-       The comparisons and the reproducibility of the methods are lacking. No table/figure had been provided dealing with the statistical comparisons of the methods. 

Authors:  Comparison and reproducibility were not analyzabled because STIR sequence is the first cross-sectional imaging competent for visualization of pelvic varicosities. US is very operator-dependant and phlebography is invasive and has never been able to realize an entire pelvic vein cartography.

  The retrospective nature of this paper is a limitation. Please discuss such a point in a dedicated limitation section.

Authors: The retrospective review of imaging examinations could lead to recognition of records.

    Acronyms should be expressed at their first mention both in the abstract and in the main text. Please revise the entire paper.

Authors: We add accordingly.

Reviewer 2 Report

1. Although informed consent was waived, the patients should have approved prior to/during their hospitalization to usage of their medical records (some kind of informed consent should have been signed).

2. The statistical data represented in the results section has not been represented at all in table/figure form.

3. The first part of the objective stated at the end of the introduction is misleading:" The aim of this study was to evaluate indirect criteria of PVI of STIR sequence retrospectively comparing with phleb-
ographic findings."

4. It  is inconclusive from the material and method section what this "indirect criteria of PVI of STIR sequence are".

5. Discussion section should provide a comparative assessment of other studies, while also referring to the current study.

6. Some non-academic phrases and English language errors require corrections.

Author Response

The statistical data represented in the results section has not been represented at all in table/figure form

Authors: The Table 2 was added accordingly.

Other results are not relevant in a table or a figure.

Although informed consent was waived, the patients should have approved prior to/during their hospitalization to usage of their medical records (some kind of informed consent should have been signed).

Authors: IRB was obtained.

The first part of the objective stated at the end of the introduction is misleading:" The aim of this study was to evaluate indirect criteria of PVI of STIR sequence retrospectively comparing with phleb-
ographic findings."

Authors: We modified it accordingly.

4It  is inconclusive from the material and method section what this "indirect criteria of PVI of STIR sequence are".

Authors: We modified it accordingly.

5. Discussion section should provide a comparative assessment of other studies, while also referring to the current study.

Authors: no other study evaluated STIR sequences versus phlebography.

Round 2

Reviewer 2 Report

1. The tables are now missing from the pdf version which was available for download.

2. Some minor English language corrections are required.

3. The discussion section needs to be reorganized as it sums up your findings, and then refers to literature data surrounding the subject/imaging techniques. Please try to integrate your findings/approach with the information contained among the literature you have cited.

Author Response

  1. The tables are now missing from the pdf version which was available for download.

Authors : Sorry we added the table at the end of the manuscript.

  1. Some minor English language corrections are required.
  2. The discussion section needs to be reorganized as it sums up your findings, and then refers to literature data surrounding the subject/imaging techniques. Please try to integrate your findings/approach with the information contained among the literature you have cited.

Authors : We write it accordingly, and we added this 2 references:

Valero I, Garcia-Jimenez R, Valdevieso P, Garcia-Mejido JA, Gonzalez-Herráez JV, Pelayo-Delgado I,    et al. Identification of Pelvic Congestion Syndrome Using Transvaginal Ultrasonography. A Useful Tool. Tomography. 2022 Jan 4;8(1):89–99. doi: 10.3390/tomography8010008. PMID: 35076614; PMCID

  1. Gavrilov SG. Vulvar varicosities: diagnosis, treatment, and prevention. Int J Womens Health. 2017 Jun 28;9:463-475. doi: 10.2147/IJWH.S126165. PMID: 28721102; PMCID: PMC5500487.